# Data-driven exclusion criteria for instrumental variable studies

**Tony Liu**                                                                    LIUTONY@SEAS.UPENN.EDU
*University of Pennsylvania, Roblox*

**Patrick Lawlor**                                                              LAWLORP1@CHOP.EDU
*The Children's Hospital of Philadelphia*

**Lyle Ungar**                                                                  UNGAR@CIS.UPENN.EDU
*University of Pennsylvania*

**Konrad Kording**                                                             KOERDING@GMAIL.COM
*University of Pennsylvania*

**Editors:** Bernhard Schölkopf, Caroline Uhler and Kun Zhang

## Abstract

When using instrumental variables for causal inference, it is common practice to apply specific exclusion criteria to the data prior to estimation. This exclusion, critical for study design, is often done in an ad hoc manner, informed by a priori hypotheses and domain knowledge. In this study, we frame exclusion as a data-driven estimation problem, and apply flexible machine learning methods to estimate the probability of a unit complying with the instrument. We demonstrate how excluding likely noncompliers can increase power while maintaining valid treatment effect estimates. We show the utility of our approach with a fuzzy regression discontinuity analysis of the effect of initial diabetes diagnosis on follow-up blood sugar levels. Data-driven exclusion criterion can help improve both power and external validity for various quasi-experimental settings.

**Keywords:** instrumental variables, exclusion criteria, compliance estimation, fuzzy regression discontinuity design, clinical guidelines, medical claims data, diabetes diagnostic criteria

## 1. Introduction

Instrumental variable (IV) analysis[1] is a popular method for evaluating causal effects from observational data. An observable variable (instrument, $Z$) is unconfounded, affects treatment $T$, and thus provides effective randomization to estimate the causal effect of treatment on the outcome $Y$. An important step, explicitly when running an IV study, or implicit when analyzing existing data, is deciding which samples to exclude. For example, suppose we want to evaluate the effectiveness of cancer screening using clinical guideline screening ages as an instrument (Kadiyala and Strumpf (2016)). We may only wish to study women because that is our target population of interest, e.g. we are considering breast cancer screening guidelines that only apply to women. Alternatively, we may believe a priori women are more likely to be *compliant* with the cancer screening guidelines, e.g. women are more likely to engage in preventative care (Brett and Burt (2001)). In the latter case, we are defining exclusion criteria as a way to increase the possibility of our study succeeding: we want to maximize our statistical power.

Practitioners often seek these data situations where the candidate instrument $Z$ is highly correlated with the treatment of interest $T$, so-called *strong instruments*. It is thus also desirable to screen

---

1. We want to congratulate Joshua Angrist, David Card, and Guido Imbens for the 2021 Nobel Prize in Economic Sciences for work, among many other contributions, on this class of quasi-experimental methods.

for weak instruments and strengthen them by excluding individuals with certain characteristics that are not compliant with the instrument (e.g. men do not adhere to cancer clinical guidelines). However, this screening must be done in a principled manner, as ad hoc analyses can introduce bias and statistical size distortions in the treatment effect estimates (Swerdlow et al. (2016); Andrews et al. (2019)). In our cancer screening example, if the decision to exclude men was made *after* we observed in the data that women had higher rates of compliance to the cancer screening instrument, we would be "data dredging" in a sense for a sample where our instrument is strong. In order to apply exclusion criteria to increase power while also maintaining valid estimates, we need an algorithmic procedure that uses the relationship between instrument compliance and individual characteristics in an honest way. Furthermore, in complex domains human experts may be limited in their ability to understand how an individual's covariates relate to compliance, making it attractive to find a data-driven method for exclusion.

Here we explore a data-driven framework for excluding samples from an IV analysis. We first examine the relationship between compliance status and statistical power of an IV study, illustrating a tradeoff between total sample size and the exclusion of individuals who are unlikely to comply. We then propose a procedure to perform IV analysis, which involves training a model to learn the probability of compliance and excluding individuals that do not meet a threshold of predicted compliance as defined by the data. We use sample splitting to ensure *honest* estimates (Athey and Imbens (2016)) and show that our treatment effect estimates are valid. We explore the performance of our method in simulated data, illustrating the benefits of data-driven exclusion on a study's power. We also present a detailed case study of our method using real-world medical claims data to evaluate diabetes diagnostic criteria through a fuzzy regression discontinuity design. We conclude with discussions on the interpretation of treatment effect estimates made with our method as well as scenarios where data-driven exclusion criteria may be most useful.

## 1.1. Related work and our contribution

Estimation of compliance status has been explored in a number of contexts. Aronow and Carnegie (2013) propose parametric methods of estimating the probability of compliance and show how the compliance score can be used in a weighting scheme to estimate the average treatment effect under certain assumptions. Joffe et al. (2003) consider using the probability of compliance in randomized trials to make correct treatment effect inferences in the presence of non-compliance. Li and Pearl (2019) formalize learning compliance status using counterfactual logic. In particularly relevant work, Huntington-Klein (2020) and Coussens and Spiess (2021) also consider compliance status estimation as a well-posed machine learning problem, using estimated compliance in a weighting approaches to improve IV inference. However, as soft weighting approaches up-weight units more likely to be compliant with continuous coefficients, it is more difficult to get a handle on internal validity: what are the characteristics of the likely compliers in the analysis sample?

Here we study the use of predicted compliance to inform exclusion criteria, a hard 0-1 weighting scheme as opposed to a soft continuous weighting scheme. Though soft weighting schemes may be more efficient from a strictly statistical viewpoint, by focusing on hard exclusion we can use machine learning to not only help with (1) more efficient treatment effect estimates but also with (2) characterizing which populations the treatment effect estimates apply to. By determining which individuals are included or excluded from our analysis sample, we can provide both an improvement in study power and a greater degree of internal validity.

Another body of literature also considers hard exclusion to improve instrumental variable analysis through a "near-far" matching strategy, which pairs units with similar covariates but different instrument levels to exclude units that may weaken the instrument (Baiocchi et al. (2010); Keele et al. (2016); Heng et al. (2020)). However, both the matching and soft-weighting approaches make it hard to define external validity: who will the treatment estimation results generalize to? Here we take a different approach by training machine learning models to predict compliance status and subsequently define exclusion criteria. With trained compliance models, we could also characterize previously unseen individuals as a "likely" complier or non-complier. We thus obtain a more meaningful way of defining external validity for the resulting treatment effect estimates. Therefore, the technique we describe here may be a promising way to aid individualized decision making, where the results of our exclusion procedure could also define whether the treatment effect estimate is potentially applicable to a particular individual.

## 2. Instrumental variable preliminaries

Here we briefly review the IV analysis framework which will frame our subsequent discussion. We use standard potential outcomes notation (Imbens and Rubin (2015)), where for an individual $i$ we have the binary instrument $Z_i$, the outcome of interest $Y_i$, the potential outcomes $Y_i(\cdot)$, the binary treatment assignment $T_i$, the potential treatments $T_i(\cdot)$, and their pre-treatment covariates $\boldsymbol{X}_i$.

We also define compliance status: *compliers* are individuals such that $T_i(Z_i) = Z_i$, namely that they receive the treatment when above the threshold and do not receive treatment when below the threshold, while *never-takers* never receive treatment ($T_i(\cdot) = 0$) and *always-takers* always receive treatment ($T_i(\cdot) = 1$). Both never-takers and always-takers are non-compliant individuals. We operate under the standard assumptions for valid IV analysis with two-sided noncompliance, namely: *relevance*, *instrument randomization*, *exclusion restriction*, and *monotonicity*. We refer the reader to e.g. Baiocchi et al. (2014) or Imbens (2014) for a comprehensive discussion on IV validity. Under these assumptions, the treatment effect estimate can be interpreted as the average treatment effect for compliers, known as the local average treatment effect (LATE):

$$\tau_{\text{LATE}} = E[Y(1) - Y(0)|T(1) = 1, T(0) = 0] \tag{1}$$
$$= E[Y(1) - Y(0)| \text{ compliers}]$$

For our subsequent $\tau_{\text{LATE}}$ estimation we will use the standard two-stage least squares (TSLS) framework (see e.g. Imbens (2014); Angrist and Pischke (2008) for details on IV effect estimation).

## 3. Statistical power and compliance

Here we study the relationship between compliance status, treatment effect estimator variance, and power. From Freeman et al. (2013) and under homoskedastic noise, we have the following asymptotic power function for an $\alpha$-level two-sided hypothesis test for true treatment effect $\tau_{\text{LATE}}$:

$$\beta(\tau_{\text{LATE}}) = 1 + \Phi\left( -\frac{\tau_{\text{LATE}}}{\sqrt{V_{IV}}} - z_{\alpha/2} \right) - \Phi\left( -\frac{\tau_{\text{LATE}}}{\sqrt{V_{IV}}} + z_{\alpha/2} \right) \tag{2}$$

where $z_t$ is the $t$th percentile Normal distribution PPF, $V_{IV}$ is the asymptotic variance of our treatment effect estimate $\hat{\tau}$, and $\Phi$ is the Normal distribution CDF. For a fixed true treatment $\tau_{\text{LATE}}$,

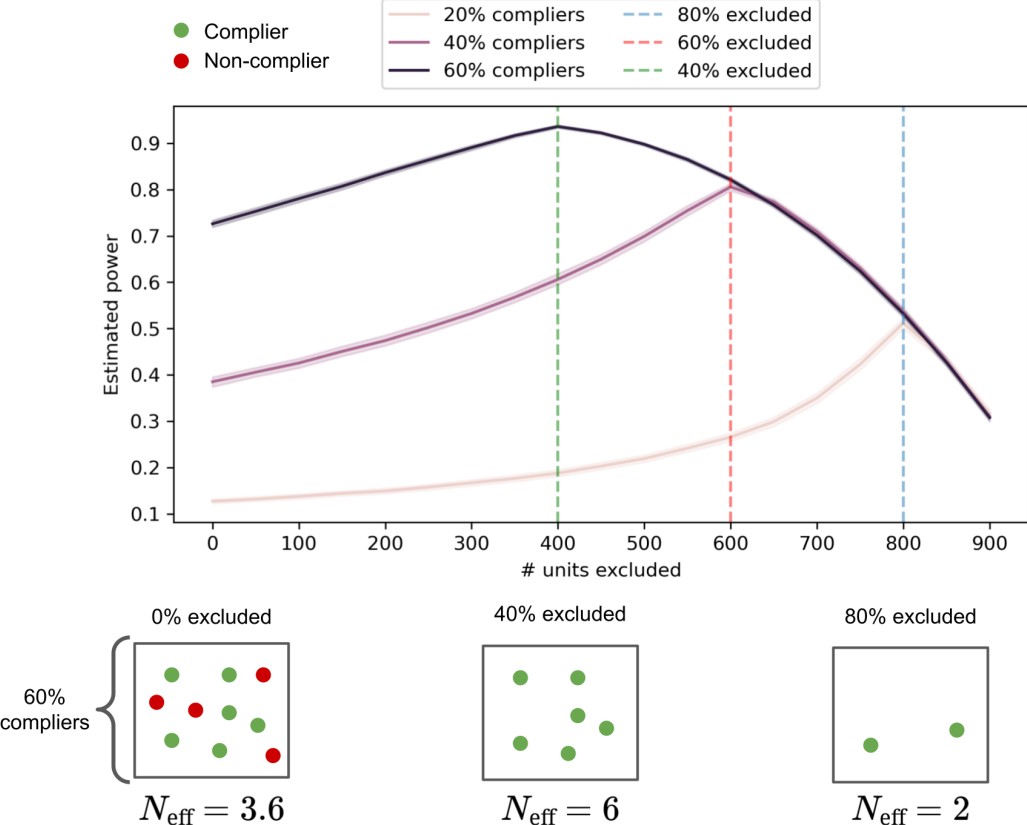

Figure 1: **Excluding non-compliers improves power.** Power calculations for simulated IV data over 100 trials (above) and a pictorial representation of the 60% complier case (below). Shaded regions are 95% confidence intervals. An $n = 1000$ sample is generated for each trial and fraction complier setting.

the power of an IV analysis can be estimated by using the sample estimator of treatment effect variance, $\hat{V}_{IV}^2$. Thus, in order to maximize the statistical power we need to minimize the IV variance.

To motivate exclusion based on compliance, we show in Appendix A.1 that removing known non-compliant units *decreases* IV variance despite reducing overall sample size, provided that the proportion of compliers $p_{\text{comply}}$ is non-zero. It should be noted however that in order to avoid potential bias introduced by excluding only known non-compliant units with observed $T_i \neq Z_i$, so-called per-protocol analysis (Imbens and Rubin (2015); Hernán and Robins (2020)), we need a method of identifying compliance status generally. If we additionally assume constant treatment effects, the IV variance can be expressed as follows (see Baiocchi et al. (2014) and Appendix A.2):

$$V_{IV}^2 = \frac{Var[Y|Z, \text{compliers}]}{Np_{\text{comply}}^2 E[Z](1 - E[Z])} \tag{3}$$

Where $N$ is the total number of units in the sample. From the denominator of this expression we can see that the IV variance of the full sample $N$ is equivalent to the variance of having $N \times p_{\text{comply}}^2$

samples. We can think of this quantity as the "effective" sample size (Heng et al. (2020)):

$$N_{\text{eff}} = N \times p_{\text{comply}}^2 \tag{4}$$

Thus, the removal of all non-compliers, if correctly identified, will actually increase the effective sample size of a study to $N \times p_{\text{comply}}$ (Coussens and Spiess (2021)), reducing variance and improving power. We note that in practice there will be a trade-off based on the likelihood of a unit being a complier, but we work in the simplified case of perfect compliance knowledge here. These results are sensible as IV analyses estimate the treatment effect for compliers, so we can think of non-compliant units as only contributing noise to the treatment effect estimate.

We illustrate the relationship between excluding non-compliers and IV power through simulated data (details can be found in Appendix C.1). We consider a case where compliance and non-compliance with the treatment threshold is a pre-determined binary indicator for each unit. For our simulation, we vary the number of units excluded across three different proportions of compliers, excluding non-compliers first: $p_{\text{comply}} = \{0.2, 0.4, 0.6\}$. This mimics the infeasible ideal scenario where we have perfect knowledge of which individuals are non-compliant. We then calculate the statistical power for the subsetted dataset, using TSLS for estimating treatment effects and the power calculation with $\tau = 0.5$ (Equation 2) to match our generated data's true treatment effect for compliers. Our simulation shows the benefits of excluding non-compliers (Figure 1), as exclusion improves power up until all non-compliers have been removed from the sample.

## 4. Data-driven exclusion criteria

Now that we have illustrated how excluding non-compliant units can improve the statistical power of the study, we propose a data-driven method to determine which units to exclude.

### 4.1. Learning compliance

We would like to use information about compliance status to inform our exclusion criteria. In our previous simulation, units were excluded based on perfect knowledge of compliance status. As compliance status in practice is latent, we treat compliance status as an unknown that can be estimated from pre-treatment covariates $\mathbf{X}$. Under standard instrumental variable assumptions (Section 2), Kennedy et al. (2020) among others have shown that the probability of compliance given pre-treatment covariates is:

$$P(\text{complier}|\mathbf{X}) = E[T|\mathbf{X}, Z = 1] - E[T|\mathbf{X}, Z = 0] \tag{5}$$

Because the instrument $Z$ is unconfounded under IV assumptions, the right hand side of Equation 5 can be viewed analogously to conditional average treatment effect estimation. Thus any algorithm that estimates conditional average treatment effects, e.g. causal forests (Athey et al. (2019)), can be used to predict the probability of compliance without needing to specify the functional relationship between the treatment, outcomes, and covariates, where we are estimating the "treatment effect" of $Z$ on the "outcome" $T$ given covariates $\mathbf{X}$. We can use sample splitting to ensure a given sample's treatment status is not used for both model fitting and estimation, which would bias the causal estimates. Within the instrumental variable framework, compliance estimation is feasible and can leverage machine learning methods developed for conditional average treatment effect estimation.

## 4.2. Exclusion based on predicted compliance

We use the trained model of compliance as an exclusion classifier, where units with lower predicted compliance are excluded from analysis. Under the standard IV assumptions of monotonicity and instrument randomization (Section 2), the proportion of compliers in any sample is:

$$p_{\text{comply}} = E[T|Z = 1] - E[T|Z = 0] \tag{6}$$

We can obtain an unbiased estimate $\hat{p}_{\text{comply}}$ for a given sample by computing the empirical conditional means (Imbens (2014)). Thus, a candidate exclusion threshold choice $\gamma$ would be the threshold value that excludes the bottom $(1 - \hat{p}_{\text{comply}})\%$ of compliance scores. However, this choice of threshold does not take into account how well the trained model can predict compliance. Optimizing solely for $\hat{p}_{\text{comply}}$ is also problematic, as this does not consider the overall sample size.

In order to balance both how well our model's compliance prediction performance and selected sample size, we choose the exclusion threshold that maximizes the effective sample size $N_{\text{eff}}$ (Equation 4). Under constant treatment effects and homoscedasticity, $N_{\text{eff}}$ is the true effective sample size, following from Equation 3. Heng et al. (2020) show that in more general cases (specifically, when the outcome variance of non-compliers is greater than the outcome variance of compliers) this is an upper bound on the true effective sample, making it a sensible albeit optimistic value to maximize. We can choose an exclusion threshold $\gamma$ that maximizes out-of-sample $N_{\text{eff}}$ through a cross-validated hyperparameter search. This data-driven selection of $\gamma$ allows our procedure to adjust to the relative performance of the compliance prediction models.

To demonstrate the validity of our exclusion approach, we present a graphical argument (see Appendix B.1 for further discussion of how exclusion produces consistent treatment effect estimates). Under a typical IV structure (Figure 2a), $Z$ is conditionally ignorable given observable covariates $X$, $Z$ only affects $Y$ through $T$, and unmeasured confounding variables $U$ only affect $T$ and $Y$. After splitting the data into two sets, training and estimation, we learn a compliance probability estimator $f(X)$ using the training set, which yields the following $I_\gamma(X)$ classifier, indexed by a particular exclusion threshold $\gamma$:

$$I_\gamma(X_i) = \mathbf{1}[f(X_i) > \gamma] \tag{7}$$

We then obtain the modified IV graph shown in Figure 2b, defining modified $Z^*, T^*, Y^*$, where $A_i^*$ for the variable $A_i$ is defined as $A_i^* = I_\gamma(X_i)A_i + (1 - I_\gamma(X_i))\overline{A}$, and where $\overline{(\cdot)}$ indicates the sample mean of the variable among samples where $I_\gamma(X_i) = 1$. By setting variables to this sample mean, we effectively exclude these samples from the IV estimation. Since $I_\gamma(X)$ is observed and causally upstream of $Z^*, T^*, Y^*$, we still maintain a valid IV design, albeit on a different subsample of individuals than the original IV design shown in Figure 2a. The resulting treatment effect estimates will thus apply to individuals who we expect to be compliers.

We note that because the data-driven exclusion procedure defines a hard threshold for exclusion based on the the predicted probability of compliance, this method in general is not as statistically efficient compared to a soft weighting by the predicted probability of compliance (see Appendix B.2 for details). However, there is value in not only improving the efficiency of treatment effect estimates (which our method does do by choosing the cutoff threshold that maximizes $N_{\text{eff}}$) but also by being able to characterize the population of the resulting treatment effect estimates. By defining exclusion criteria in terms of a hard cutoff, practitioners can examine and characterize the sample populations of "likely" compliers and "likely" non-compliers by their covariate composition.

Thus, we can frame this choice in terms of trading off efficiency for interpretability: practitioners may rather have an estimator that sufficiently improves efficiency while maintaining interpretability, rather than a less interpretable estimator that is more precisely efficient.

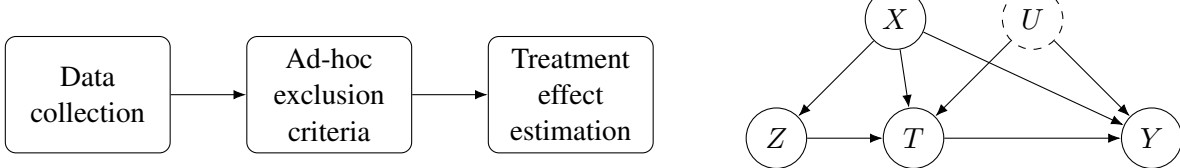

(a) Typical instrumental variable study (left) and corresponding graph (right).

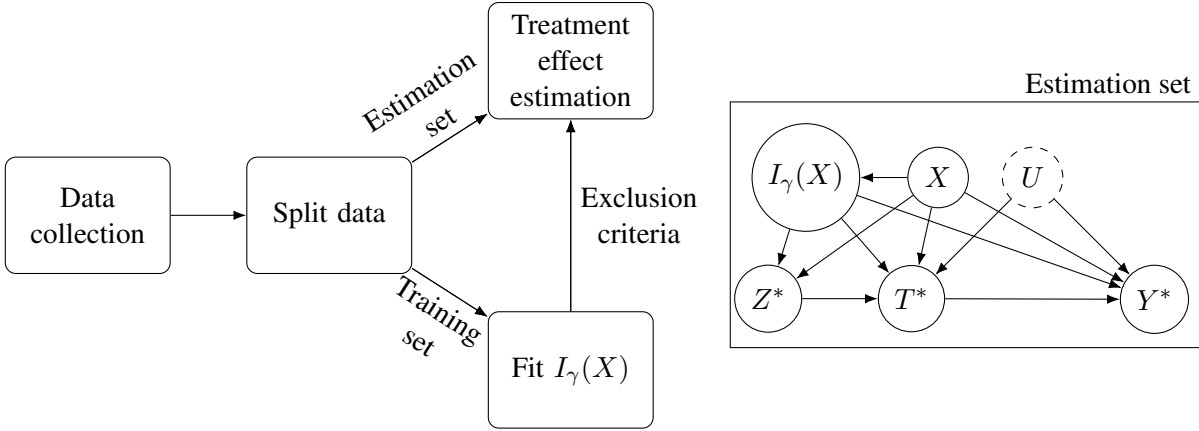

(b) Data-driven exclusion instrumental variable study (left) and corresponding graph (right).

Figure 2: **Data-driven exclusion treatment estimates are valid through sample splitting.** Applying the exclusion classifier $I_\gamma(X)$ to a held-out estimation set is like controlling for another pre-treatment covariate.

### 4.3. Data-driven exclusion procedure

Given IV data $(T_i, Z_i, Y_i, \boldsymbol{X}_i)_{i=1}^n$, a completely data-driven method for determining exclusion criteria and performing analysis is as follows:

1. Partition the data into two parts, $S_1$ and $S_2$.

2. Use $S_1$ to fit a compliance model $f$ using pre-treatment covariates $\boldsymbol{X}$ (Equation 5).

3. Use the fitted model to predict probability of compliance for units in $S_2$.

4. Select exclusion threshold $\gamma$ that maximizes out-of-sample $N_{\text{eff}}$ through a $k$-fold cross-validation search using $S_1$, with $N_{\text{eff}}$ evaluated per fold for a grid of candidate $\gamma$ values.

5. Define exclusion classifier $I_{\gamma, S_1}(\boldsymbol{X})$ (Equation 7) and use it to exclude units from $S_2$. Call remaining subset $S_2'$.

6. Swap the roles of $S_1$ and $S_2$ in steps 1-5 to produce a subset $S_1'$.

7. Perform TSLS estimation on the "cross-fitted" sample $S_1' \cup S_2'$.

Because we have a hard threshold for exclusion, we are able to define two disjoint sets of individuals that can be combined for downstream treatment effect estimation. Each individual is only used for compliance prediction once, which allows for the entire data to be considered while also preventing overfitting when training the compliance models (Athey and Imbens (2016)). This approach can also easily be extended to a k-fold setting. By using the observed relationship between compliance and pre-treatment covariates, we have a data-driven method for determining exclusion criteria that will improve the precision of our treatment effect estimates.

### 4.4. Data-driven exclusion simulation and results

We use simulated data to evaluate the effectiveness of our method in increasing IV study power[2]. Here, we generate five pre-treatment covariates for each individual ($\boldsymbol{X_i} = \{X_{i1}, ..., X_{i5}\}$) that also influence the probability of compliance. We use causal forests to fit the compliance probability estimates in step 2 of our procedure. We evaluate three methods for estimating the treatment effect across our simulations: simple TSLS analysis (no additional covariates) using the full data, TSLS analysis that includes the pre-treatment covariates using the full data, and our data-driven exclusion criteria method presented above, using simple TSLS analysis for estimating treatment effects on the data partition $S_1' \cup S_2'$. We compare these methods across three different proportions of compliers: $p_{\text{comply}} = \{0.2, 0.4, 0.6\}$ as well as three covariate settings where $\boldsymbol{X_i}$ (1) strongly predicts compliance status, (2) weakly predicts compliance status, and (3) does not predict compliance status. Further details about the data generation process for our simulations can be found in Appendix C.2.

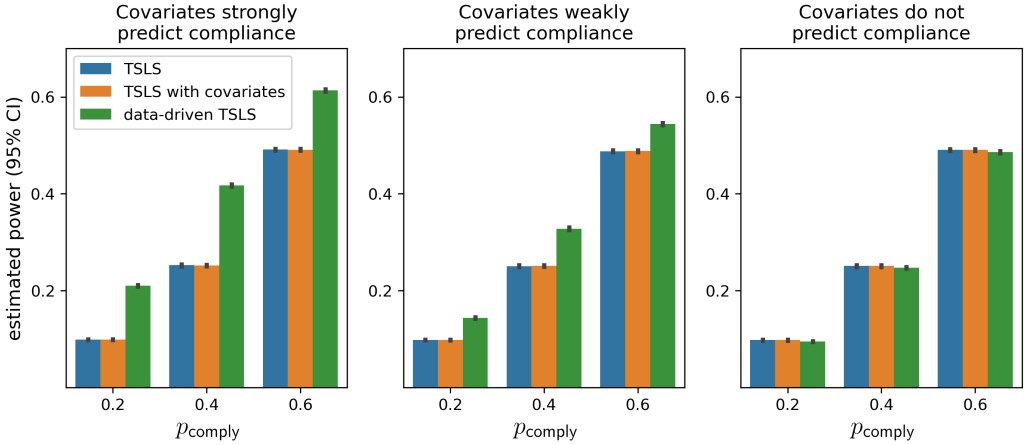

Figure 3: **Data-driven exclusion improves power when compliance can be predicted.** Estimated power for simulated IV data with pre-treatment covariates influencing compliance over 500 trials, with error bars as 95% confidence intervals. A $n = 2000$ sample is generated for each trial.

Our empirical simulation demonstrates the benefits of our data-driven exclusion criterion (Figure 3). Across all settings of $p_{\text{comply}}$ and compliance prediction strength, the treatment effect estimates are unbiased for the three methods we consider (see Figure C.1 for corresponding estimated treatment effect plots). Our data-driven exclusion method produces the highest power across all $p_{\text{comply}}$ cases when the covariates $\boldsymbol{X}$ either strongly or weakly predict compliance, with larger power

---

2. Source code can be found at: https://github.com/tliu526/data-driven-exclusion

gains when compliance status can be strongly predicted. Importantly, in the case when covariates do not predict compliance, our method still produces study power comparable to the baseline TSLS methods. Our simulation results suggest that our data-driven method for exclusion could be useful in situations where there are observed pre-treatment covariates that may predict compliance well.

To give a sense of how our exclusion threshold optimization based on out-of-sample $N_{\text{eff}}$ works in practice, we report the mean number of individuals excluded for the $p_{\text{comply}} = 0.6$ case of our simulation (see Table C.2 for all cases). Across the "strongly predicted compliance," "weakly predicted compliance," and "unpredictable compliance" cases, 749 units (37.5% of total sample), 560 units (28.0% of total sample), and 22 units (1.1% of total sample) were excluded, respectively. We see in the "strongly predicted compliance" case, our classifiers exclude nearly the same proportion of units (37.5%) as the actual proportion of non-compliers (40%), producing a substantial power improvement (Figure 3, left). In the "weakly predicted compliance" case, our classifiers are more conservative on average with 28.0% of the sample excluded, which still provides a modest power benefit (Figure 3, center). Finally, in the "unpredictable compliance" case, our classifiers do not exclude many units at all (1.1%), showing no power benefit but also no substantial decrease (Figure 3, right). These empirical results illustrate how our method for selecting the $\gamma$ threshold can adjust to the relative performance of the trained compliance models in practice.

## 5. Case study: evaluating clinical guidelines using insurance claims data

To explore the effectiveness of our data-driven exclusion method on real-world data, we use a fuzzy regression discontinuity design (FRDD) for estimating how clinical guidelines for type II diabetes diagnosis affect subsequent blood sugar levels – how much does initial diagnosis and treatment help follow-up blood sugar control? We also present this case study as a model for future health policy evaluation: guidelines are critical in clinical decision making but may not evaluated experimentally, and often lend themselves to quasi-experimental strategies like regression discontinuity designs due to treatment decisions being made on a continuous variable cutoff, such as age or a lab test.

### 5.1. Data and causal identification strategy

In order to evaluate how type II diabetes diagnoses affects follow-up blood sugar, we use data from Optum's de-identified Clinformatics® Data Mart Database (2001-2016), which contains medical claims, prescriptions, and lab results for patients covered under commercial health plans and Medicare. Patients are filtered for no prior diabetes diagnosis or diabetes medication prescription. Pre-treatment covariates we use for compliance estimation include age, race, gender, insurance type, socio-economic factors, and initial encounter date (see Appendix D.1 for further data details).

We examine the use of A1C as a diagnostic criteria for diabetes under a fuzzy regression discontinuity design (FRDD). A1C is a marker of long-term blood sugar and by American Diabetic Association guidelines set in 2010, an A1C of $\geq 6.5\%$ is the basis for type 2 diabetes (American Diabetes Association (2010)); patients above this threshold should be diagnosed with diabetes. We use a 30 day window from the initial A1C reading to detect the presence or absence of diabetes diagnoses as treatment $T$ and the first A1C reading at least a year after the initial reading as the follow-up outcome $Y$. This outcome measure is intended to capture how the diabetes diagnosis and subsequent treatment (such as diet changes, exercise encouragement, or medication prescription) affects a patient's follow-up blood sugar levels. We find a discontinuous jump in type II diabetes diagnosis rates at the 6.5% threshold, evidence that the FRDD strategy may be fruitful (Figure 4).

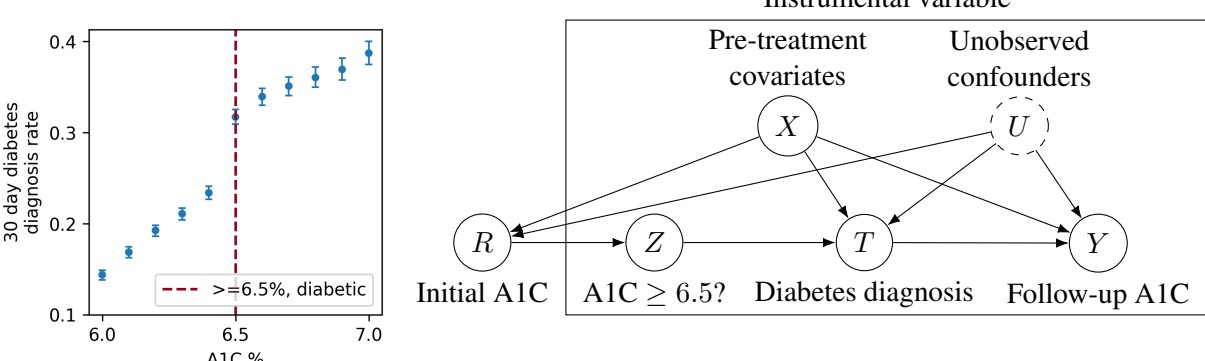

Figure 4: **There is a clear discontinuity in diabetes diagnosis rate at the 6.5% A1C threshold (left), which allows for a fuzzy regression discontinuity (FRDD) approach (right).** Shown left are diabetes diagnosis rates across A1C levels, with 95% confidence intervals as error bars. Shown right are the causal relations between the relevant variables; the FRDD approach resembles an IV.

As treatment effect estimation in an FRDD is numerically equivalent to two-stage least squares (Imbens and Lemieux (2007), also see Appendix D.2 for treatment effect estimation details) we can use our data-driven exclusion method (Section 4.3) to evaluate this FRDD.

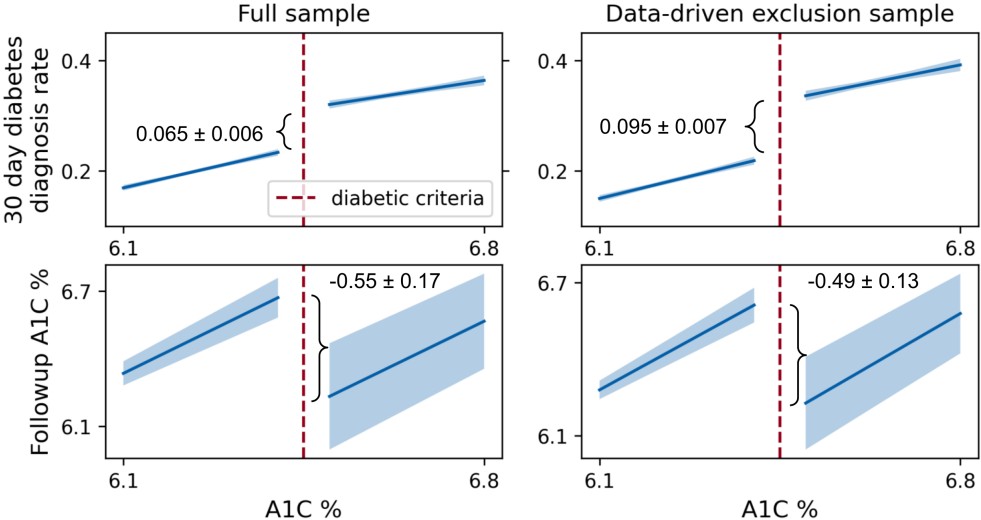

Figure 5: **Data-driven exclusion strengthens the first-stage regression and tightens confidence intervals in the outcome regression.** Top row shows the first stage regression (diabetes diagnosis probability) while the bottom row shows the outcome regression (follow-up A1C), with 95% CIs plotted in the shaded region.

## 5.2. Results

Applying data-driven exclusion to our sample improves first-stage instrument strength, (Figure 5) as we see a larger jump in treatment probability at the 6.5% threshold when comparing our data-driven

exclusion sample ($0.095, se = 0.007$) to the original sample ($0.065, se = 0.006$). Even though 32,641 out of the original 96,361 samples are excluded, $N_{\text{eff}}$ actually increases from 1,889 to 2,015.

|  | Full sample | Data-driven exclusion sample |
|---|---|---|
| $N$ | 96,361 | 63,720 |
| $N_{\text{eff}}$ | 1,889 | 2,015 |
| Treatment effect (SE) | -0.55 (0.17) | -0.49 (0.13) |
| Treatment effect CI | (-0.88, -0.22) | (-0.75, -0.24) |

Table 1: **Data-driven exclusion improves estimated treatment effect precision.**

We see that a diabetes diagnosis at the 6.5% A1C threshold decreases follow-up A1C% levels by roughly 0.5% (Table 1). Both the full sample and data-driven exclusion sample treatment effect estimates are within each other's confidence intervals. However, the exclusion sample's confidence interval (CI length 0.51) is shorter than the full sample's confidence interval (CI length 0.66), indicating increased precision in the exclusion sample. Furthermore, at a nominal $\tau = -0.5$ level (Equation 2), the full sample power is $\beta_{\text{full}}(-0.5) = 0.84$ while the exclusion sample power is $\beta_{\text{excl}}(-0.5) = 0.97$. This case study demonstrates how our data-driven exclusion method can improve internal validity and feasibility of quasi-experimental studies.

### 5.3. Interpretability and validity

Our method also allows for improved external validity and interpretation. Because we use a hard compliance score threshold for excluding samples, we can characterize who is likely to be compliant and non-compliant according to the trained classifiers (Table 2). We see that individuals who are included are more likely to be older, female, and on Medicare as opposed to commercial insurance. Most notably, individuals who are included have their initial A1C reading much later in time (average date 02/2013) than individuals who were excluded (average date 11/2005). These results align with our expectations of compliance status – the 6.5% A1C threshold was not officially established as a diagnostic criteria for diabetes until 2010 (American Diabetes Association (2010)) so more compliant individuals should have initial encounter dates after 2010, while it has been shown individuals on Medicare have greater healthcare utilization for preventative care and follow-up treatment (Card et al. (2008)).

Furthermore, we can use our trained exclusion classifiers $I_{\gamma,S_1}(X), I_{\gamma,S_2}(X)$ to decide whether a previously unseen individual $i$ would be included or excluded in the analysis sample: if both classifiers predict that $i$ should be included, then $i$ would be characterized as a "likely" complier. This could easily be extended to a $k$-voting scheme among classifiers $I_{\gamma,k}(X)$ if we use $k$-fold sample splitting. As an aside, we could also characterize likely compliers by applying feature importance interpretability methods to our trained models (e.g. SHAP values by Lundberg and Lee (2017)), which we leave for future work.

### 5.4. Case study limitations

It is also important to highlight the limitations of this study design. We have verified the correctness of the causal graph shown in Figure 4 in consultation with clinical domain experts. However, there

| Covariate | Included | Excluded |
|---|---|---|
| Number of samples | 63,720 | 32,641 |
| Avg. age (SD) | 62.9 (12.5) | 56.0 (11.1) |
| Female | 53.3% | 50.4% |
| On Medicare | 54.4% | 21.1% |
| Avg. month/year encounter date | 02/2013 | 11/2005 |

Table 2: **We can characterize likely (non-) compliers by examining who is excluded or included.**

are additional factors that warrant further investigation that could improve our internal validity as a clinical finding. In particular, this dataset was chosen to facilitate the evaluation of our data-driven exclusion method as opposed to maximizing study validity (see Appendix D.3 for further discussion), and we will continue to work with physicians to refine our study approach.

More generally, we also use medical claims data, which may suffer from selection biases in healthcare utilization (Tyree et al. (2006)) or potential mismatches between claims reporting of diabetes diagnoses and true patient status (Lin et al. (2005)). Though the quasi-experimental FRDD strategy can safeguard against unmeasured confounding, we plan on conducting a follow-up study with different data sources in order to validate these results.

## 6. Discussion

Here we have explored the relationship between compliance, the use of compliance estimates in conducting IV studies, and the statistical power of IV studies. We illustrate how pre-treatment covariates can be used to build models of compliance probability, and how we can use estimates of the overall compliance rate and effective sample size with these models to maximize power. We provide a data-driven procedure that uses these models for determining exclusion criteria, and apply this method to both simulated and real-world data. Our data-driven exclusion approach not only improves our estimates of the treatment effect but also provides a method which can determine to whom these estimates apply. We now discuss the interpretation of the resulting estimates, consider scenarios where our method is likely to be beneficial, as well as raise limitations and future work.

### 6.1. Interpretation

We recognize that our data-driven exclusion procedure may increase the difficulty of interpreting the resulting treatment effect estimates. By selectively excluding units estimated to be non-compliant, we are estimating the average treatment effect on "likely" compliers, which is not strictly the local average treatment effect. In reality however, many IV studies are implicitly making their treatment estimates on expected compliers as well. Recalling our cancer screening example from Section 1, investigators who exclude males from their study are conducting analyses on a sample they believe to be more compliant with the treatment intervention. Since this is often done in an ad-hoc manner based on domain knowledge, our method is useful in providing an objective way to determine exclusion criteria.

Furthermore, because we choose to use a hard exclusion threshold for compliance classification, we can characterize the likely complier population sample which would aid interpretation in practice. This is in contrast to other soft weighting methods that may be more statistically efficient

but less interpretable (Section 4.2). For example, if a doctor wanted to know whether the reported treatment estimates in our diabetes case study applied to a particular patient, they could input the patient's covariate profile into the exclusion classifiers to see if they would have been excluded in the study: if they were not included, then the treatment estimates would likely not be applicable to them. Thus, our method could also facilitate validation of any resulting estimates with domain experts.

## 6.2. Limitations and future work

We highlight some limitations and opportunities for future work. First, our method hinges on whether compliance can be predicted from observed covariates. We can empirically evaluate the quality of this prediction by examining the chosen exclusion threshold $\gamma$ of our classifiers as well as the number of units being excluded – if compliance cannot be predicted well, then a low $\gamma$ value will be chosen and few units will be excluded, akin to the compliance estimation case we considered in our simulation (rightmost plot, Figure 3).

In cases where compliance can be predicted however, our method is likely to be useful. One domain where we believe this holds true is in health policy (Moscoe et al. (2015)). Instrument compliance may depend on observable covariates such as insurance type, socioeconomic status, and previous medical history, and we demonstrate the feasibility of our method in this domain with our diabetes case study. We believe that quasi-experimental strategies used in conjunction with large medical datasets show promise in evaluating clinical guidelines and could even inform future work on how they can be improved (Marinescu et al. (2019)).

Another limitation is that our methodology for choosing the number of units to exclude (steps 4-5 in Section 4.3) is only a heuristic based on the estimated proportion of compliers $\hat{p}_{\text{comply}}$ and effective sample size $N_{\text{eff}}$, though we demonstrate its usefulness in practice. Given the tradeoff between sample size and non-compliance we explored in Section 3, there may be an *optimal* threshold to use for exclusion that takes the uncertainty of the predicted compliance probability into account. Future work could explore the relationship between the errors of the machine learning model (Yadlowsky et al. (2021)) and estimates of what the exclusion threshold should be in order to not only optimize power but other utility functions, e.g. the monetary cost of including a sample.

## 6.3. Conclusion

Deciding which samples to exclude in an IV study is a critical processing step, yet is often performed in an ad-hoc manner based on an investigator's domain knowledge. The data-driven procedure we have proposed provides an alternative, objective method for determining such exclusion criteria. Our method increases internal validity by excluding subjects that would weaken the instrument as well as external validity by providing classifiers that determine who the results may apply to. Data-driven exclusion criteria can improve both the power and interpretation of IV studies, improving feasibility and allowing for quasi-experimental studies in a wider range of data domains.

## Acknowledgments

We thank Rahul Ladhania, Luke Keele, Ben Lansdell, Ari Benjamin, and Brian Collopy for their discussions and feedback which have improved this work.

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

## Appendix A. Proofs for compliance status and variance

### A.1. Removing known non-compliers reduces variance

We show that removing known non-complying individuals (with observed $T_i \neq Z_i$) increases statistical power, provided the compliance rate is greater than 0. As discussed in Section 3, maximizing the statistical power is equivalent to minimizing the IV estimator variance. Under the assumption of homoscedastic noise, the variance of the IV estimator $\hat{\tau}$ is proportional to the following (Angrist and Imbens (1995)):

$$\hat{V}_{IV}^2 \propto \frac{Var(Z)}{Cov^2(T,Z)} \tag{8}$$

We give the following lemmas for variance and covariance calculations in binary variables:

**Lemma 1** *Given a sample of size $n$ of a binary variable X, let $k$ be the number of samples where $X = 1$. The variance estimate of $X$ is then:*

$$\frac{k(n-k)}{n(n-1)}$$

**Lemma 2** *Given a sample of size $n$ of pairs $(x_i, y_i)$ of binary variables X and Y, let $k_X$ be the number of samples where $X = 1$, let $k_Y$ be the number of samples where $Y = 1$, and let $k_{XY}$ be the number of samples where $X = Y = 1$. The estimated covariance between X and Y is then:*

$$\frac{n k_{XY} - k_X k_Y}{n(n-1)}$$

We now consider a population of size $n$, where each individual $i$ has both a treatment assignment $Z_i$ and an indicator for actual treatment uptake $T_i$. From Lemmas 1 and 2, we can then write the IV variance as being proportional to the following:

$$n(n-1)\frac{k_Z(n-k_Z)}{(nk_{TZ} - k_T k_Z)^2} \tag{9}$$

We now show under what conditions removing noncompliers from our sample ($T_i \neq Z_i$) reduces the IV variance.

**Case 1: remove** $(T_i = 1, Z_i = 0)$. By removing this sample, the IV variance is now:

$$(n-1)(n-2)\frac{k_Z((n-1) - k_Z)}{((n-1)k_{TZ} - (k_T - 1)k_Z)^2} \tag{10}$$

where $k_Z$, $k_T$, and $k_{TZ}$ are the counts in the original sample, and $n$ is the original sample size. Applying some inequalities that hold because $n > 1$:

$$(n-1)(n-2)\frac{k_Z((n-1) - k_Z)}{((n-1)k_{TZ} - (k_T - 1)k_Z)^2} < n(n-1)\frac{k_Z((n-1) - k_Z)}{((n-1)k_{TZ} - (k_T - 1)k_Z)^2}$$

$$n(n-1)\frac{k_Z((n-1) - k_Z)}{(nk_{TZ} - k_T k_Z + k_Z - k_{TZ})^2} < n(n-1)\frac{k_Z(n - k_Z)}{(nk_{TZ} - k_T k_Z + k_Z - k_{TZ})^2}$$

The final term looks like the original IV variance, with an additional $k_Z - k_{TZ}$ in the denominator. Since $k_Z \geq k_{TZ}$, this difference is positive. Thus the denominator will be larger (and therefore the IV variance will be smaller) if $nk_{TZ} - k_T k_Z$ is positive, or equivalently:

$$nk_{TZ} > k_T k_Z \tag{11}$$

**Case 2: remove** $(T_i = 0, Z_i = 1)$. By removing this sample, the IV variance is now:

$$(n-1)(n-2)\frac{(k_Z-1)((n-1)-(k_Z-1))}{((n-1)k_{TZ}-k_T(k_Z-1))^2} \tag{12}$$

Again applying inequalities:

$$(n-1)(n-2)\frac{(k_Z-1)(n-k_Z)}{((n-1)k_{TZ}-k_T(k_Z-1))^2} < n(n-1)\frac{(k_Z-1)(n-k_Z)}{((n-1)k_{TZ}-k_T(k_Z-1))^2}$$

$$n(n-1)\frac{(k_Z-1)(n-k_Z)}{((nk_{TZ}-k_T k_Z+k_T-k_{TZ})^2} < n(n-1)\frac{k_Z(n-k_Z)}{((nk_{TZ}-k_T k_Z+k_T-k_{TZ})^2}$$

This again looks like the original IV variance, with the additional $k_T - k_{TZ}$ in the denominator. Since $k_T \geq k_{TZ}$, this difference is positive. So we again have the same situation where the IV variance will be smaller if:

$$nk_{TZ} > k_T k_Z$$

We then can analyze the composition of the $k_T, k_Z$, and $k_{TZ}$ terms.

Table A.1: Categorization of units based on treatment and instrument.

| $T_i$ | $Z_i$ | Unit category |
|---|---|---|
| 0 | 0 | never-taker or complier |
| 0 | 1 | never-taker |
| 1 | 0 | always-taker |
| 1 | 1 | always-taker or complier |

As shown in Table A.1, there are four possible settings of $(T_i, Z_i)$ pairs. Let $c_{tz}$ be the number of individuals in our sample where $T_i = t, Z_i = z$. Then $c_{00} + c_{11} + c_{01} + c_{10} = n$.

We then have that removing a non-complier from the sample decreases the IV variance if:

$$nk_{TZ} > k_T k_Z$$
$$c_{11}(c_{00} + c_{11} + c_{01} + c_{10}) > (c_{11} + c_{10})(c_{11} + c_{01})$$
$$c_{11}c_{00} + c_{11}^2 + c_{11}c_{01} + c_{11}c_{10} > c_{11}^2 + c_{11}c_{01} + c_{11}c_{10} + c_{10}c_{01}$$
$$c_{11}c_{00} > c_{01}c_{10} \tag{13}$$

Note here that if we continue to remove non-compliers from the sample, the RHS term will continue to decrease, also decreasing variance.

Under the *instrument randomization* assumption (Section 2), the proportion of each compliance type is the same whether $Z = 0$ or $Z = 1$. As shown in Imbens (2014), we can use this to express $c_{ij}$ in terms of the proportions of compliers ($\pi_C$), always-takers ($\pi_A$), and never-takers ($\pi_N$):

$$\pi_A = Pr(T = 1|Z = 0) = \frac{c_{10}}{c_{10} + c_{00}}$$
$$\pi_N = Pr(T = 0|Z = 1) = \frac{c_{01}}{c_{01} + c_{11}}$$
$$\pi_C = 1 - \pi_A - \pi_N$$

Expressions for $c_{11}$, $c_{00}$:

$$\frac{c_{11}}{c_{01} + c_{11}} = 1 - \pi_N = \pi_A + \pi_C$$
$$\frac{c_{00}}{c_{10} + c_{00}} = 1 - \pi_A = \pi_N + \pi_C$$

We can now revisit inequality 13:

$$c_{11}c_{00} > c_{01}c_{10}$$
$$(c_{10} + c_{00})(c_{01} + c_{11})(\pi_A + \pi_C)(\pi_N + \pi_C) > (c_{10} + c_{00})(c_{01} + c_{11})\pi_A\pi_N$$
$$(\pi_A + \pi_C)(\pi_N + \pi_C) > \pi_A\pi_N$$
$$\pi_C^2 + \pi_C\pi_A + \pi_C\pi_N + \pi_A\pi_N > \pi_A\pi_N$$
$$\pi_C(\pi_A + \pi_N + \pi_C) > 0$$
$$\pi_C > 0$$

Thus, as long as we have a non-zero proportion of compliers in our sample, removing an observed non-complier will decrease the IV estimator variance. ∎

### A.2. IV variance as a function of compliance status

Following arguments presented by Baiocchi et al. (2010) and using Theorem 3 of Angrist and Imbens (1995), with homoscedasticity and constant treatment effects we have:

$$V_{IV}^2 = \frac{Var[Y|Z, \text{compliers}]Var(Z)}{NCov^2(T, Z)} \tag{14}$$

As $Z$ is binary, $Var(Z) = E[Z](1 - E[Z])$. We write $Cov(T, Z)$ as follows:

$$
\begin{aligned}
Cov(T,Z) &= E[TZ] - E[Z]E[T] \\
&= (E[Z]E[TZ|Z=1] + (1-E[Z])E[TZ|Z=0]) - E[Z]E[T] \\
&= E[Z]E[T|Z=1] - E[Z]E[T] \\
&= E[Z]E[T|Z=1] - E[Z](E[Z]E[T|Z=1] - (1-E[Z])E[T|Z=0]) \\
&= (E[Z] - E[Z]^2)E[T|Z=1] - (E[Z] - E[Z]^2)E[T|Z=0] \\
&= E[Z](1-E[Z])(E[T|Z=1] - E[T|Z=0] \\
&= E[Z](1-E[Z])p_{\text{comply}}, \text{ from Equation 5}
\end{aligned}
\tag{15}
$$

Then from Equation 15 we have:

$$
V_{IV}^2 = \frac{Var[Y|Z, \text{compliers}]}{Np_{\text{comply}}^2 E[Z](1-E[Z])}
\tag{16}
$$

as desired. ∎

## Appendix B. Consistency and efficiency of treatment effect estimation with exclusion

### B.1. Consistency of compliance-weighted estimator

Coussens and Spiess (2021) show that there is an equivalence between weighted instruments, which our method is a variant of, and certain interacted instruments (Proposition 3 in their work). Specifically, they give:

$$
\tau_w = \frac{E[w(X)\tau_{\text{LATE}}(X)| \text{ complier}]}{E[w(X)| \text{ complier}]}
\tag{17}
$$

where $\tau_{\text{LATE}}(X)$ is the conditional local average treatment effect, akin to Equation 1:

$$
\tau_{\text{LATE}}(X) = E[Y(1) - Y(0)| \text{ complier}, X]
\tag{18}
$$

Note that $\tau_w$ is generally not equivalent to $\tau_{\text{LATE}}(X)$ unless we have constant treatment effects and homoscedasticity. They also show that given $w(X) = P(\text{complier}|X)$ (Equation 5) this weighted local average treatment effect $\tau_w$ (termed "super local average treatment effect" in Huntington-Klein (2020)) can be consistently estimated with predicted compliance score weights $\hat{w}$ under cross-fitted estimates (Proposition 7 in their work). As our exclusion classifier $I_\gamma(X_i)$ is a binarization of the compliance score $w(X_i)$ at a fixed threshold $\gamma$, these consistency results also apply to our procedure described in Section 4.3.

### B.2. Efficiency of compliance-weighted estimator

Since binarization does discard information about predicted compliance, our choice of weights will not be as efficient as weighting by compliance score, except in the limiting cases where compliance is perfectly predictable: $P(\text{complier}|X_i) \in \{0, 1\}$ or completely unpredictable: $P(\text{complier}|X_i) = p_{\text{comply}}$ (Proposition 2 of Coussens and Spiess (2021)). However, among estimators that do perform hard thresholding, our method aligns with prior work's recommendations of how to best define

the inclusion sample. Our data-driven exclusion method is akin to estimation of treatment effects of covariate-identifiable subgroups (what we call "likely" compliers) presented in Kennedy et al. (2020). Theorem 5 of Kennedy et al. (2020) shows consistent and efficient estimation via efficient influence functions of the bounds of such treatment effects under margin and accurate estimation conditions, which is equivalent to our estimation of treatment effects for included "likely" compliers:

$$E[Y(1) - Y(0)|I_\gamma(X) = 1]$$

From these results, they recommend estimating treatment effects for the $t$ individuals in the sample with the highest compliance scores, where t is some fixed minimum sample size. This is because the bounds of the "likely" complier treatment effect is minimized when selecting such a subgroup of size $t$ (Proposition 3 of Kennedy et al. (2020)). In a similar fashion, our method selects a particular proportion of the individuals with the highest compliance scores according to the threshold $\gamma$ (which also controls the minimum sample size $t$) in a data-driven manner so as to maximize the effective sample size $N_{\text{eff}}$ and statistical power (Section 4.2). Thus, our method aligns with these recommendations, while providing a data-driven method for constructing the inclusion sample in practice.

## Appendix C. Simulation details

In all simulations we present, data are generated in the following manner. Let $\pi_C, \pi_A, \pi_N$ be the proportions of compliers, always-takers, and never-takers respectively. We vary $\pi_C$ for our simulations (Figures 1 and 3) and set the number of always-takers and never-takers to the same proportion: $\pi_A = \pi_N = (1 - \pi_C)/2$. We then generate instruments $Z$, treatment $T$, and confounders $C$ from a mean zero Gaussian with $\rho_{ZT} = 0.8$, $\rho_{TC} = 0.8$, $\rho_{ZC} = 0$. For compliers, the treatment effect is $\tau$:

$$Y_i = \tau T_i + C_i + \epsilon_i \qquad (19)$$

where $\epsilon \sim \mathcal{N}(0, 1)$, while never-takers and always-takers have a treatment effect of $\tau + 0.25$ to simulate heterogeneous treatment effects.

### C.1. Binary indicator of compliance

We generate compliance indicators $X_i$ randomly for each sample such that the fraction of compliers in the data matches $\pi_C$. The covariate $X_i$ is thus a perfect predictor of compliance. For the simulation shown in Figure 1, we set $\tau = 0.5$ and evaluate the estimated power at the same level.

### C.2. Covariates that predict compliance

When we generate data to test our compliance score learning procedure, we first generate a random linear regression with five nonzero regressors $X_{i1}, ..., X_{i5}$, producing outcome $r_i$ using scikit-learn's make_regression() method (Pedregosa et al. (2011)). $r_i$ is min-max scaled to fall within the range $[0, 1]$. The bottom $\pi_N \times 100$ percentile of $r_i$ are never-takers, the middle $\pi_A \times 100$ percentile of $r_i$ are always-takers, and the top $\pi_C \times 100$ percentile of $r_i$ are compliers.

For the simulation shown in Figure 3, we set treatment effect $\tau = 0.25$, with estimated power evaluated at the same level. For the case where covariates $X$ only weakly predict compliance, we apply $\mathcal{N}(0, 75)$ Gaussian noise to the regression output. For the case where covariates do not predict compliance, $X$ do not predict $r_i$. Figure C.1 shows the treatment effect plot that corresponds to the Figure 3 simulation, where our data-driven exclusion method makes unbiased estimates of the treatment effect across difference proportion of compliers. Table C.2 shows the mean number of samples excluded for each simulation scenario using data-driven method for selecting $\gamma$ as described in Section 4.3.

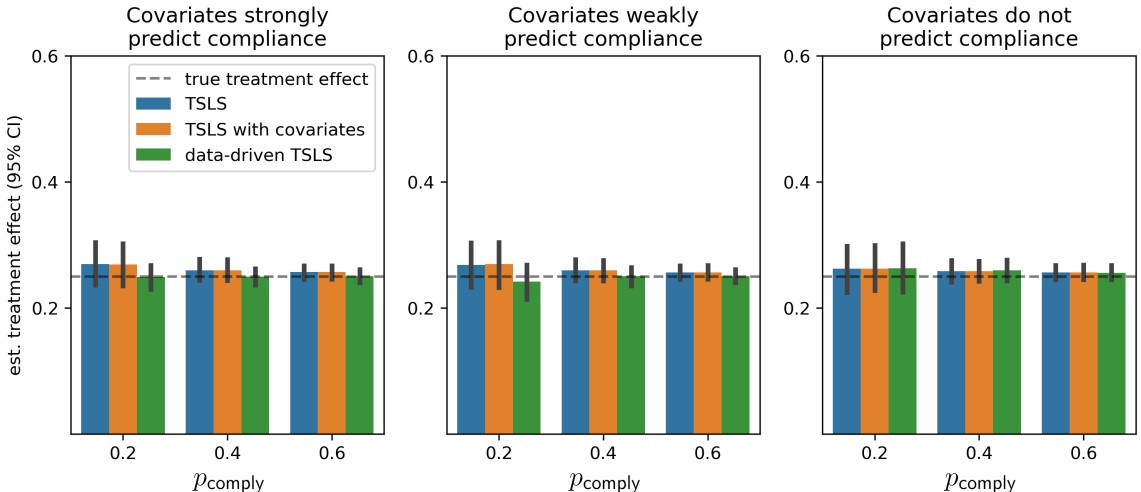

Figure C.1: **Data-driven exclusion does not bias estimates.** Estimated treatment effects for simulated IV data with pre-treatment covariates influencing probability of compliance over 500 trials. Bar plots show the mean with error bars as 95% confidence intervals. An $n = 2000$ sample is generated for each trial.

| $p_{comply}$ | Mean number of samples excluded (% of total sample) | | |
| --- | --- | --- | --- |
| | Compliance not predicted | Compliance weakly predicted | Compliance strongly predicted |
| 0.2 | 110 (5.52%) | 1379 (68.95%) | 1572 (78.61%) |
| 0.4 | 36 (1.8%) | 955 (47.75%) | 1169 (58.47%) |
| 0.6 | 23 (1.14%) | 560 (27.97%) | 749 (37.47%) |

Table C.2: **Data-driven exclusion adapts the number of samples excluded (via the threshold $\gamma$) to how well compliance can be predicted.**

## Appendix D. Diabetes FRDD study details

### D.1. Medical claims dataset

Insurance claims were pulled from a database containing both commercial (COM) and Medicare (MDC) insurance plans from January, 2001 to December 2016. We index an individual by their first recorded medical encounter with an A1C reading, so as to capture the initial blood test for diabetes. The dataset consists of individuals who:

- had no diagnosis of Type II diabetes prior to the first encounter date

- had no prescription of blood sugar controlling medication prior to the first encounter

- have a recorded age and gender with their insurance plan

- are at least 18 years old at the time of the first encounter

All filters on diagnosis codes and medication prescriptions were based on the presence of corresponding International Classification of Diseases (ICD-9) and National Drug Code (NDC) codes in their insurance claims. We use a 30 day window for defining initial diabetes diagnosis because of temporal lags in how insurance claims are filed.

Our sample thus has individuals with no observed history of diabetes diagnosis or treatment so as to capture their (potentially) initial diagnosis. We then define our outcome measure as a follow-up A1C recorded in the database, at least a year but no longer than two years from the initial encounter date. Pre-treatment covariates include age, race, insurance status, encounter date, and numerous socio-economic factors: net worth, household income, education level, home ownership, and a federal poverty line indicator. In total, there are 284,996 individuals within the database that make up our study sample.

### D.2. Treatment effect estimation

As shown by Imbens and Lemieux (2007), a fuzzy regression discontinuity design with a rectangular kernel within the bandwidth for local linear regression can be framed as a TSLS estimation problem with the additional regressors:

$$\begin{pmatrix} 1 \\ \mathbf{1}[R_i < c](R_i - c) \\ \mathbf{1}[R_i \geq c](R_i - c) \end{pmatrix} \tag{20}$$

Where $R_i$ is the running variable (in our case, initial A1C reading), and $c$ is the treatment threshold cutoff (A1C = 6.5%). Note that the instrument $Z_i$ is defined as $Z_i = \mathbf{1}[R_i \geq c]$. We use Imbens-Kalyanaraman optimal bandwidth selection to select the bandwidth of analysis for local linear regression (Imbens and Kalyanaraman (2009)), yielding a bandwidth of 0.36. As A1C readings are rounded to the first decimal place, we include 6.1 in the lower local regression so the lower and upper regressions have the same number of A1C levels as support. A1C ranges from [6.1, 6.5) make up the lower local regression, while A1C ranges from [6.5, 6.8] make up the upper local regression.

We use all covariates discussed in Appendix D.1 for compliance prediction, which include missing indicator columns for the socio-economic factors. We follow the data-driven exclusion procedure we presented in Section 4.3, using a two-fold sample split, causal forests (Athey et al. (2019)) for compliance estimation, and five-fold nested cross-validation for determining exclusion threshold $\gamma$.

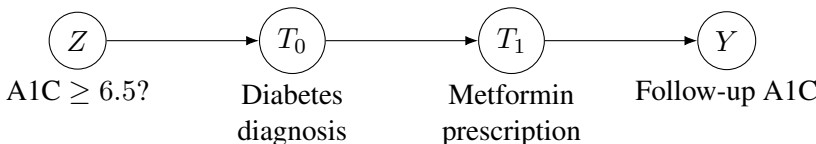

Figure D.2: **Subgraph of treatment for future study refinement.**

### D.3. Future study refinement

As noted in Section 5.3, there are opportunities to further refine our case study as a clinical result. Because our goal with this work is to evaluate how well data-driven exclusion works in practice with real data, we included initial encounter date as well as insurance type as covariates for compliance prediction. To improve study design validity, we will consider defining the sample a priori as individuals who had encounter dates after the A1C guideline was officially in place only on one type of insurance when presenting our results as a clinical finding.

Additionally, the causal graph shown in Figure 4, while correct to the best of our knowledge, can be further refined as well. We use diabetes diagnosis as our "treatment" of interest, as we presume that following this diagnosis on a patient's initial encounter, their physician would then recommend a number of treatment options, such as lifestyle changes (diet and exercise) or even the prescription of blood sugar control medication. We note that our finding of a ∼0.5% reduction of follow-up A1C levels is sensible, in roughly the same vicinity of the effect size of exercise (0.3 - 0.9%), diet changes (0.5 - 2.0%) (American Diabetes Association (2017a)), and metformin (0.9 - 1.1%) (American Diabetes Association (2017b)) has on follow-up A1C levels. Because of these different treatment paths, there are intermediate nodes in between treatment and outcome that could be considered to make the treatment of interest more precise, such as by using metformin refills, a commonly prescribed drug for initial diabetes control (Figure D.2).

Additionally, there are fuzzy regression discontinuity feasibility tests that need to be considered to ensure the correctness of the clinical finding. As a falsification test, we plot the density of the running variable, which shows no sign of potential manipulation at the 6.5% threshold (Figure D.3), but additional falsification tests such as inspecting the continuity of the pre-treatment covariates at the threshold should be considered in more detail (Bor et al. (2014)) to increase the study validity.

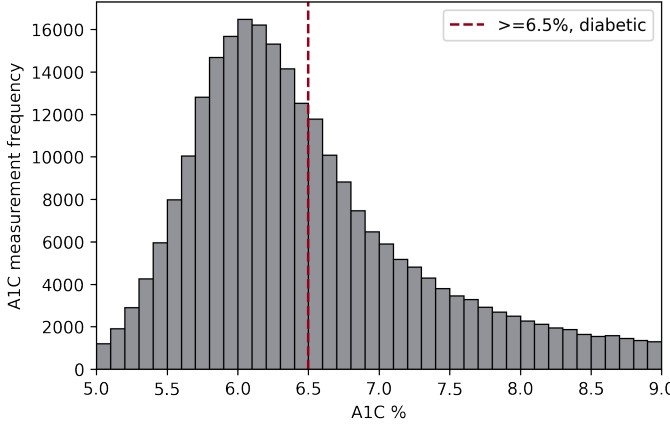

Figure D.3: **Running variable density plot shows no visual signs of "bunching" at the threshold, which would have been an indicator of potential manipulation of the treatment assignment.**

