# OpenReview forum: "Data-driven exclusion criteria for instrumental variable studies"
_cclear.cc/CLeaR/2022/Conference — CLeaR 2022 Poster_

### Official Review · Reviewer_KAnD · 2021-11-22

**Confidence:** 4
**Overall Score:** 3

**Main Review:**

Originality: Using predicted non-compliance status as an exclusion criterion is novel to me, as far as I am aware. The paper presents a new method. It’s not clear to me how this method is justified compared to simply excluded the observed non-compliers, especially considering it requires a supervised-learning step to train a classifier, so it in fact requires the it already be known who is and is not a complier. It’s unclear to me also how this procedure is related to issuing instrumental variables in particular: non-compliers are a problem for estimating treatment effect whenever we aren’t satisfied with simply estimating the effect of intent to treat.

Significance: Removing sources of noise or error from causal inference analysis is an important problem, and non-compliers are certainly one such problem. It’s not clear to me that this procedure is much more advanced than simply removing the non-compliers though, which is an obvious point of comparison. I don’t feel that this offers new insights particularly, since it’s’ just building a compliance classifier and excluding people who score above a threshold on it. I don’t expect it to have broad impacts outside the community unless it can be thoroughly demonstrated that this approach is better than all other ways of dealing with non-compliance. Being able to predict whether someone is likely to comply with medication or not is of course useful in a clinical setting, but what is being offered in this paper is not a well trained and evaluated compliance prediction model for some treatment, but putatively a method for improving performance of instrumental variable based causal inference.

Technical quality: The approach appears technically sound to me. The claim that removing non-compliers improves power is supported. I don’t notice any technical errors, and it’s not unnecessarily complicated.

Clarity: The submission is clearly written and organized. The take home message is easy to understand even from just the abstract. The motivation for dealing with non-compliers is clear, but it’s not clear why their particular strategy is the most appropriate way to do this. The technical details are all clear.


**Summary:**

This paper presents a data driven method for identifying study participants who should be excluded in order to improve performance for instrumental variable based causal inference analysis. The excluded participants are selected based on their predicted status as a non-complier.

---

> ### Author Response · Authors · 2021-12-04
> **Response to Reviewer KAnD**
>
> Thank you for the detailed comments, as they highlight important points of comparison for our method. However, we believe that the reservations expressed come from some misunderstanding of our problem framing, and we apologize for any lack of clarity in our text. We respond to specific points inline below and will clarify our submission considerably if given the opportunity to revise it.
>
> >It’s not clear to me how this method is justified compared to simply excluded the observed non-compliers...
>
> Excluding observed non-compliers ($T_i \neq Z_i$) is an option; this is equivalent to per-protocol analysis sometimes used in medicine. However, this can introduce bias into the treatment effect estimate, as there will be a mixture of both compliers and non-compliers when $T_i = Z_i$ (Table 3; also see Chapter 23.9 Imbens and Rubin 2015). Per-protocol analyses often require adjusting for confounding and selection bias in order to be valid (Chapter 9.5 Hernán and Robins 2020). As one motivation in using IV analysis is to avoid the unconfoundedness assumption, we generally cannot exclude the observed non-compliers and have our identification assumptions hold. Thus we have to consider other methods for non-complier removal.
>
> For a point of comparison, we ran a per-protocol exclusion strategy in the “strongly predicted compliance” scenario presented in Section 4.4. The true treatment effect is 0.25, with both our data-driven method and standard IV analysis showing little to no bias (Figure C.6, left). However, excluding observed non-compliers introduces substantial bias, with mean treatment effect estimates [95% CI] of 0.42 [0.41, 0.43], 0.36 [0.35, 0.37], 0.31 [0.31, 0.32] for compliance proportions of 0.2, 0.4, 0.6 respectively. This highlights the need to explicitly model non-compliers as we cannot simply remove observed non-compliers for valid estimation. We will discuss this point and provide additional simulation results if given an opportunity to submit a revision.
>
> >...especially considering it requires a supervised-learning step to train a classifier, so it in fact requires it already be known who is and is not a complier.
>
> Our method actually does not require knowledge of compliance status, but rather relies on standard IV assumptions. Compliance status is not generally observed, but under positivity, relevance, instrument randomization, and monotonicity assumptions, the probability of compliance given covariates $X$ can be estimated (Kennedy et al. 2020). This is shown in Equation 5 of our submission, where the probability of compliance given covariates can be estimated using observed values of $T, Z, X$.
>
> As discussed in Section 4.1, this problem can be viewed analogously to conditional average treatment effect (CATE) estimation, and so compliance probability can be predicted using the same ML methods for CATE estimation despite not having direct knowledge of compliance status.
>
> >It’s unclear to me also how this procedure is related to issuing instrumental variables in particular...
>
> Our goal in IV analysis is to make treatment effect estimates beyond the intent to treat effect. It has been shown that under standard IV assumptions, in particular monotonicity, the treatment effect estimate made by IV analyses can be interpreted as the local average treatment effect, also sometimes called the complier average causal effect (Equation 1, also see Angrist and Imbens 1995, Chapter 24.5 Imbens and Rubin 2015).
>
> Within this instrumental variables framework, the treatment effect of interest *is* the average treatment effect for compliers, and so it is desirable to remove non-compliers. We also benchmark our method against standard IVs in both simulated (Figure C.6) and real-world (Figure 5) settings, and show that the resulting estimates are comparable.
>
> >I don’t feel that this offers new insights particularly...I don’t expect it to have broad impacts outside the community unless it can be thoroughly demonstrated that this approach is better than all other ways of dealing with non-compliance.
>
> As noted at the beginning of our comment, we feel that this conclusion comes from some misunderstanding of our text, and we again apologize for any lack of clarity. We hope that our responses have helped address the concerns raised.
>
> Our contribution to the community is primarily the introduction and exploration of a new logical setting for causal inference. By splitting IV analysis into two problems, 1) “who should be included in the analysis?” and 2) “what is their resulting causal estimate?”, as well as proposing a data-driven solution to 1), we can improve statistical power while also providing interpretability of the sample of interest (Table 2). Thus, in our opinion this work is a unique contribution to the literature. Importantly, it combines causal validity with statistical power improvement in a way that lends itself to real-world medical applications, a direction we are  pursuing with our clinical collaborators.

---

### Official Review · Reviewer_VPyL · 2021-11-22

**Confidence:** 3
**Overall Score:** 7

**Main Review:**

This paper is well-organized and clearly written. The contribution of the paper is very clear and the limitations are also discussed by the authors. The main contributions are:

1. The authors proved that excluding non-compliers (assuming that we know who are non-compliers and who are not) can improve the power for the test for $\tau_{LATE}$

2. The authors provided a method to estimate the conditional probability of a unit to be a complier, given the pretreatment covariates $\bf X$, and provide a heuristic data-driven exclusion procedure to exclude the units with a low conditional probability to be a complier.

3. The proposed method was applied to a real-world problem, and the result seems convincing.

One limitation of the work, as mentioned by the authors in Section 6.2, is that the method depends on whether compliance can be predicted from observed covariates. The authors mentioned in Section 6.2 that “we can empirically evaluate the quality of this prediction by examining the chosen exclusion threshold of our classifiers as well as the number of units being excluded”, which sounds interesting to me. Intuitively, I can imagine that if the compliance status is totally unpredictable, then the learned classifier will return $f(X_i)=0.5$ (approximately) for every unit $i$, meaning that no sample should be dropped, since otherwise $N_{eff}$ will drop. Except for this extreme case, I would like to know whether this rule can be applied in the situation that $X_i$ can predict compliance but the model is misspecified. Besides, it would be better to report the details about how $\gamma$ is selected and compare the selected $\gamma$’s in the three scenarios in Section 4.4.


**Summary:**

This paper provides data-driven exclusion criteria to exclude non-compliers in instrumental variable studies.

---

> ### Author Response · Authors · 2021-12-04
> **Response to Reviewer VPyL**
>
> Thank you for the constructive feedback. We address individual comments inline below:
>
> > Intuitively, I can imagine that if the compliance status is totally unpredictable...no sample should be dropped, since otherwise $N_{eff}$ will drop. Except for this extreme case, I would like to know whether this rule can be applied in the situation that can predict compliance but the model is misspecified.
>
> Yes, our method can work in situations even when the strength of compliance prediction varies. We show this in our simulation results presented in Figure 3: the right-most plot shows the exact case raised where compliance status is completely unpredictable, while the middle plot shows the scenario where there is a weak signal in the pre-treatment covariates for compliance prediction: there are still some power improvements over standard IV analysis but the gains are more modest than when the covariates strongly predict compliance (Figure 3, left). And because we consider compliance modeling as solely a predictive task, the trained ML models are robust to model misspecification, akin to how many methods for conditional average treatment effect (CATE) estimation do not require the specification of the functional relationship between the treatment effects and covariate profile of individuals (e.g. causal trees, Athey and Imbens 2016). We discuss this connection between CATE estimation and compliance prediction in Section 4.1.
>
> > Besides, it would be better to report the details about how gamma is selected and compare the selected gamma’s in the three scenarios in Section 4.4.
>
> We would be happy to provide more details on how $\gamma$ is selected if given the opportunity to submit a revision. To elaborate further on the cross-validated hyperparameter search we discuss in Section 4.3 (step 4): within each partition of the data $S_i$, we conduct a $K$-fold cross validation search over an evenly spaced array of potential $\gamma$ values given a compliance model trained on $K-1$ folds, with $n_\text{eff}$ evaluated for each $\gamma$ value on the held-out fold. The $\gamma$ with the best mean $n_{eff}$ across the $K$ splits is chosen. For the particular simulations presented in Section 4.4, $K$ is set to 5.
>
> We can also provide further details on the average number of individuals excluded by our compliance classifiers across the Section 4.4 scenarios in a potential revision (note that we can also provide the selected $\gamma$ values, but the key metric for comparison is the number of individuals excluded across the simulation settings since the value of $\gamma$ will be somewhat dependent on the particular compliance model trained for a given trial). To give a sense of these results, we’ve calculated the mean number of individuals excluded for the $p_\text{comply} = 0.6$ case of our simulation ($n=2000$, 500 trials) across the three cases of “strongly predicted compliance,” “weakly predicted compliance,” and “unpredictable compliance” respectively: 749 units excluded (37.5% of total sample), 559 excluded (28.0% of total sample), and 22 units excluded (1.1% of total sample). We see in the “strongly predicted compliance” case, our classifiers exclude nearly the same proportion of units (37.5%) as the actual proportion of non-compliers (40%). In the “weakly predicted compliance” case, our classifiers are more conservative on average with 28.0% of the sample excluded, which still provides a modest power benefit. Finally, in the “unpredictable compliance” case, our classifiers do not exclude many units at all (1.1%), showing no power benefit but also no significant decrease. These results illustrate how our method for selecting the $\gamma$ threshold can adjust to the relative performance of the trained compliance models.
>
> We also want to highlight that the data-driven selection process of $\gamma$ is a heuristic based on the estimated proportion of compilers and $N_\text{eff}$ that we have found to perform well empirically, as discussed in Section 6.2. In future work, we would like to explore optimal settings of the threshold $\gamma$ according to either the types of errors made by the compliance estimation model (Yadlowsky et al. 2021) or an alternative utility function.

---

### Official Review · Reviewer_cEUM · 2021-11-23

**Confidence:** 4
**Overall Score:** 7

**Main Review:**

This is a well written paper proposing a data-driven approach to exclude non-complier individuals for the estimation of treatment effects on an instrumental variables setup. The significance of this method is that it offers a data-driven alternative to traditional ad-hoc exclusion approaches and can be used to categorize unseen individuals for which the treatment effect can apply. This method promises to be a useful tool for big data clinical applications. In simulations, where all the necessary assumptions hold, the authors show that a) their method produced unbiased causal effects estimations under all tested conditions of percentage of compliers in the sample and prediction strength of the covariates, and b) their method showed better statistical power than 2SLS and 2SLS with covariates, for the conditions where the covariates are good predictors of the compliance (and equal power in conditions where the covariates are weak predictors of compliance). Figure 3 and C6 results suggest that the proposed method is a better alternative than standard 2SLS and 2SLS with covariates. In addition, strengthening the data-driven nature of the method, it uses a cross-validation approach to select the compliance threshold (hyperparameter) based on maximizing the size of the effective sample. Some points require clarification:

The authors claim that their method could be useful in situations where there are observed pre-treatment covariates that may predict compliance well. Is it possible to correctly determine in advance when pre-treatment covariates are good predictors of compliance? And related, how good needs to be a good predictor to be considered here? Can the authors comment more about this point, researchers would benefit from more clear guidelines about a proper selection of the pre-treatment covariate set.

Related to the above point. In the empirical application, the authors mention (in the appendix) that they use a set of covariates to predict compliance (note: there is a small discrepancy between the covariates listed in the main text and the ones in the appendix D1). Can the authors explain why they used this particular set of covariates? Do the authors have a sense of the performance of the method under different subsets of covariates? And related, is there a data-driven way to choose the best set of covariates?

In the empirical application, the authors explain that they use a fuzzy regression discontinuity design (FRDD), and then explain that treatment effect estimation in this design is numerically equivalent to 2SLS. Was the use of FRDD a decision constrained by the dataset used?

Do the authors have a sense of how sensitive is the method (in terms of power and effects estimation) to an incorrect removal of non-compliers? For example, what happens if I randomly exclude samples?

Some typos:
Figure C.6. has an incorrect label in the y-axis, and incorrect labels for the bars.
In the first sentence of Appendix A.1, the notation should be T instead of W.
In Appendix C, first paragraph, rho_TC appears two times with different values (0.8 and 0).
In Appendix C, first paragraph, pi_B should be pi_N in the proportion of compliers pi_A = pi_B = (1-pi_C)/2.



**Summary:**

The authors propose a data-driven method for excluding non-compliance samples in an instrumental variable set to compute treatment effects. The method learns the probability of compliance and excludes individuals based on a threshold. The authors illustrate their approach in simulation varying the percentage of non-compliers in the sample and the strength of pre-treatment covariates to predict compliance. The authors also apply their method to an empirical medical dataset to evaluate the effect of diabetes diagnosis on blood sugar levels. The method is based on a hard decision threshold on compliance that allows a clear characterization of the population for which the treatment effect estimates apply. In addition, given that the model is fitted with testing data, new unseen individuals can be categorized or not as likely compliers. The data-driven strategy to remove non-compliers is based on the relationship between compliance and statistical power and prediction of compliance probability from a set of pre-treatment covariates.

---

> ### Author Response · Authors · 2021-12-04
> **Response to Reviewer cEUM**
>
>
> Thank you for the positive and detailed feedback. We respond to questions inline below:
> >Is it possible to correctly determine in advance when pre-treatment covariates are good predictors of compliance?...how good needs to be a good predictor to be considered here?
>
> The general guidelines for selecting covariates are to ensure 1) the variables are in fact pre-treatment and 2) are not colliders, so as to not introduce M-bias. Given that a treatment decision can sometimes be a process rather than a point event (especially in medicine), it is critical for the practitioner to use domain knowledge/expertise to ensure that no bias is introduced. So long as these conditions are satisfied, we can frame compliance estimation as a supervised learning task, and use standard practices in machine learning feature selection. Thus, while we may only have an intuition of what covariates are good predictors of compliance in advance, the ML model can learn in a data-driven manner which pre-treatment covariates to use for compliance prediction.
>
> We also use a data-driven approach to inform “how good” the compliance predictor needs to be. As discussed in Section 4.2, we maximize out-of-sample $N_\text{eff}$ as a heuristic for selecting the exclusion threshold $\gamma$. Suppose in the suboptimal case, the trained model $f(X_i)$ cannot predict compliance better than chance. Since exclusion at any level of $\gamma$ would decrease $N_\text{eff}$, the out-of-sample selection process should choose $\gamma$ such that no individuals are excluded. Our method should then do no worse than typical IV analysis. We show this scenario in Figure 3, right; our method shows no improvement in power but importantly, no significant degradation either. Similarly, if our compliance model can improve out of sample $N_\text{eff}$ even marginally, $\gamma$ will be set appropriately to see a benefit in power (Figure 3, center).
>
> >Can the authors explain why they used this particular set of covariates? Do the authors have a sense of the performance of the method under different subsets of covariates?
>
> We chose these covariates based on guidance from our clinical collaborators to capture typical confounders in the public health space. There are three groups of factors that we identify: 1) demographics: race, age, and gender, 2) socio-economic factors (SES): education level, net worth, etc, and 3) insurance utilization: encounter date, commercial vs Medicare. Given the presumed high correlation among the SES factors, we hypothesize that only a subset are needed for good performance. We also believe that the insurance variables are the most important, as the data are derived from medical claims. It would be interesting to run analysis on subgroups of features to verify these hypotheses; ML feature importances can also be used to study which features drive performance, and we plan on conducting such analyses in the future (Section 5.3).
>
> >...is there a data-driven way to choose the best set of covariates?
>
> We refer to our first response above to address this question.
>
> >Was the use of FRDD a decision constrained by the dataset used?
>
> Correct; because the diagnostic criteria for diabetes is an A1C level above 6.5%, the instrument used for this dataset is the indicator function $Z = \mathbf{1}[ \text{A1C} \ge 6.5]$. By constructing this $Z$ and verifying that there is a discontinuous jump in treatment probability (Figure 4, left), treatment effect estimation can be approached as an IV analysis.
>
> >Do the authors have a sense of how sensitive is the method (in terms of power and effects estimation) to an incorrect removal of non-compliers? ...what happens if I randomly exclude samples?
>
> Random exclusion of samples will not bias effect estimation, but will increase variance and in turn decrease power (Equation 2). To see this, consider a scenario where we randomly remove one sample. Sample size will decrease to $N -1$, but the proportion of compilers, on average, will remain the same. Let $c$ be the number of compilers, $p = c/N$ the original proportion of compliers, and $p_{new}$ the new proportion of compliers:
>
> $$
> p_{new} = \frac{c - p}{N-1} \text{, as a random sample has probability p of being a complier}
> $$
>
> $$
> p_{new} = \frac{c}{N-1} - \frac{c / N}{N-1}  = \frac{Nc - c}{N(N-1)} = \frac{(N-1)c}{N(N-1)} = p
> $$
> Since the total sample size decreases while maintaining the same proportion of compliers on average, $N_{eff}$ will decrease (Equation 4), increasing variance and decreasing power. This matches intuition; if we take random draws of e.g. half the sample, treatment effect estimates will on average be unchanged but will also have considerably more variability. Thus, we need to ensure that any removal of samples is informed by how well compliance can be predicted, which we have discussed in Section 4.2 as well as in the comments above.
>
> >Some typos…
>
> Thank you for catching these typos; we will correct them if given the opportunity to submit a revision.

---

### Author Response · Authors · 2021-12-04
**General Response**

We thank the reviewers for their insightful comments which have helped us improve our work. If given the opportunity to revise our submission, we would make the following changes:

- Correct typos and discrepancies raised by Reviewer cUEM
- Elaborate on how the relative strength of the compliance classifier affects our method based on points raised by both Reviewers cUEM and VPyL
- Include more details on the hyperparameter search for $\gamma$ per comments raised by Reviewers VPyL and cUEM
- Include additional discussion of the strategy of removing observed non-compliers in comparison to our method per comments made by Reviewer KAnD

We want to also highlight that our main contribution is exploring new *logic* to causal inference in this domain. We consider the process of conducting an instrumental variable analysis as two separate problems: 1) determining which samples to exclude for analysis and 2) performing inference on the sample. Problem 2) is well-studied with many advances in the literature, but our point of emphasis is that problem 1) can be approached with data-driven methods as well. In typical studies, problem 1) is solved by human domain experts, often in an ad-hoc manner. By framing problem 1) as a well-posed machine learning task, we can use our data-driven exclusion approach to obtain unbiased estimates of causal effects while also improving power, provided that compliance can be estimated. Thus, we believe that this work provides a compelling contribution to the literature by opening and improving empirical quasi-experimental analysis opportunities where compliance can be predicted, as we have demonstrated in our case study using real-world medical data.

---

### Decision · Program_Chairs · 2022-01-12

**Decision:**

Accept (Poster)

**Comment:**

In this paper, the authors make an observation that estimating compliance probability for a unit may be used to improve estimation efficiency by excluding rows of data that are likely to contain non-compliers.

Overall the paper was received positively. However, one concern the authors should address is this:

Statistical theory generally suggests the most efficient estimator will not throw away data. Since the authors throw away data based on complier probability, it seems to suggest that their algorithm might be a "hard thresholding" special case of a weighted estimator that uses compliance probabilities. This estimator is likely related to the efficient influence function of the complier specific ATE parameter (which is known to attain the efficiency bound in non-parametric models). It would be good to discuss the relationship of the authors' estimator to such estimators.